# Improved DeepSORT-Based Object Tracking in Foggy Weather for AVs Using Sematic Labels and Fused Appearance Feature Network

**DOI:** 10.3390/s24144692

**Published:** 2024-07-19

**Authors:** Isaac Ogunrinde, Shonda Bernadin

**Affiliations:** Department of Electrical and Computer Engineering, FAMU-FSU College of Engineering, Tallahassee, FL 32310, USA; bernadin@eng.famu.fsu.edu

**Keywords:** multi-object tracking, DeepSORT, object detection, sensor fusion, deep learning, autonomous vehicles, radars, adverse weather, fog

## Abstract

The presence of fog in the background can prevent small and distant objects from being detected, let alone tracked. Under safety-critical conditions, multi-object tracking models require faster tracking speed while maintaining high object-tracking accuracy. The original DeepSORT algorithm used YOLOv4 for the detection phase and a simple neural network for the deep appearance descriptor. Consequently, the feature map generated loses relevant details about the track being matched with a given detection in fog. Targets with a high degree of appearance similarity on the detection frame are more likely to be mismatched, resulting in identity switches or track failures in heavy fog. We propose an improved multi-object tracking model based on the DeepSORT algorithm to improve tracking accuracy and speed under foggy weather conditions. First, we employed our camera-radar fusion network (CR-YOLOnet) in the detection phase for faster and more accurate object detection. We proposed an appearance feature network to replace the basic convolutional neural network. We incorporated GhostNet to take the place of the traditional convolutional layers to generate more features and reduce computational complexities and costs. We adopted a segmentation module and fed the semantic labels of the corresponding input frame to add rich semantic information to the low-level appearance feature maps. Our proposed method outperformed YOLOv5 + DeepSORT with a 35.15% increase in multi-object tracking accuracy, a 32.65% increase in multi-object tracking precision, a speed increase by 37.56%, and identity switches decreased by 46.81%.

## 1. Introduction

Object tracking is constantly determining a moving object’s trajectory from measurements taken by one or more sensors [1]. Single-object tracking (SOT) [2] and Multi-object tracking (MOT) [3,4,5,6,7] are two main categories of Object tracking methods (MOT). When using SOT, the tracker follows a single, predetermined object. Object tracking is required as soon as a target appears in the first frame and must be tracked in all subsequent frames. Multi-object tracking (MOT) necessitates a detection step to identify all targets of a particular class and monitor them individually without any previous information about their appearance or amount. This is a far more difficult endeavor, as a number of issues, such as object occlusion and objects with similar looks, may make tracking more difficult [1]. In object tracking, track loss occurs when false measurements are used in a tracking filter, which causes the estimation error to diverge [8].

Recently, state-of-the-art MOT research has centered on two methods: (i) tracking by detection [9,10,11,12,13,14] and (ii) joint tracking and detection [15,16]. In this work, our focus is detection by tracking. During tracking by detection, an object detector is used to detect objects in a frame and provide that detection information to the object tracking algorithm to perform the frame-to-frame association of the objects. For instance, if five objects were detected in a given frame, five distinct bounding boxes would be generated and tracked throughout all future frames. However, detecting and tracking frame-by-frame is laborious and may prevent MOT from being executed in real-time, thus reducing the level of object-tracking performance. Other challenges facing object tracking include the lack of balance between tracking speed and accuracy, background distractions, noise (such as fog) in the background, multi-spatial-scaled objects, and occlusion.

As previously mentioned, the initial step of tracking by detection algorithm is to detect the objects that need to be tracked. Autonomous vehicles (AVs) often use a variety of sensors, including cameras, lidars, and radars, to detect objects such as (pedestrians, cars, trucks, bicyclists, traffic lights and signs, etc.), in their path [17,18,19]. However, inclement weather, including heavy fog, snow, rain, and sandstorms, can drastically reduce sensor performance [20,21,22,23,24,25]. For instance, low visibility in heavy foggy weather makes it difficult for cameras to detect objects, increasing the likelihood of collisions and fatalities [24,26]. On the other hand, there is a loss of reflectivity and an inaccuracy in distance measurement when using lidar in fog. By monitoring how much energy is reflected from radio waves, radars can calculate the range and speed of an object using the doppler effect. Thus, radars outperform cameras and lidars in bad weather and remain consistent regardless of atmospheric conditions. The data from radars is too sparse for object classification due to the low density of radar point clouds [27,28]. However, AVs’ radar and camera fusion systems can provide complementary information for detected objects [19,28,29,30].

Wojke et al. proposed DeepSORT [10], which uses YOLOv4 for the detection phase. The traditional YOLOv4 model is a single-sensor system that takes only video sequences as input. The original DeepSORT is a simple neural network for deep appearance descriptors such that the feature map extracted is prone to losing relevant details about the track being matched with a given detection in fog. In heavy fog, targets with a high degree of appearance similarity on the detection frame are more likely to be mismatched with the wrong predictions, resulting in identity switches or track failures. Matching inaccuracies occur when objects of various sizes are on the same detection frame. Tracking small, colored, distant, and widely varied-sized objects can be challenging and yield inaccurate outcomes if the background is too noisy or excessively busy with objects of similar color. It is simpler for object detectors to identify and track objects with a uniform background. Therefore, an input frame containing objects with strong color contrast works best for object tracking. It is important to have a solid framework that can boost detection and tracking capabilities while decreasing the number of identity switches and tracking failures in fog. In this paper, based on the DeepSORT algorithm [10] in Figure 1, we present an improved deep learning-based multi-object tracking approach in Figure 2. We address: (i) the issues of background distractions and noise caused by fog that can cause detection and prediction mismatches; and (ii) the balance between tracking accuracy and tracking speed.

We make the following contributions to achieve improved tracking speed and tracking accuracy under foggy weather conditions:Instead of a single sensor modal system (video sequence only) used for the detection phase in [9,10], we employed our deep camera-radar fusion network (CR-YOLOnet model) [31] for faster and more accurate object detection in the detection phase of our improved deep learning-based MOT in Figure 2. Our CR-YOLOnet model reached an accuracy (mAP at 0.50) of 0.849 and a speed of 69 fps for small and distant object detection under medium and heavy fog conditions.We simulated a real-time autonomous driving environment in CARLA [32]. In addition to the radar and camera sensors, we obtain semantic labels of the ego vehicle environment using semantic segmentation cameras. The semantic segmentation camera presents each object in its field of vision with a distinct color corresponding to the predetermined object category (label). We fed the semantic labels into the segmentation module of our own deep convolutional neural network-based appearance descriptor.We replaced the basic convolutional neural network used for appearance descriptors in DeepSORT with our deep convolutional neural network. Our deep appearance descriptor uses a cross-stage partial connection (CSP)-based backbone for low-level feature extraction and a feature pyramid network (FPN)-based neck for multi-scale feature vectors to address objects of different sizes. We incorporate GhostNet into our deep appearance descriptor to replace the traditional convolutional layers used in standard neural networks. Using GhostNet helps to: (i) generate more features, thus improving the integrity of the feature extracted for an accurate detection and prediction match; (ii) reduce the number of parameters, computational complexities, and cost, thus improving tracking speed without diminishing the output feature map.We incorporate a segmentation module to add rich semantic information to the low-level appearance feature map generated using semantic labels. With semantic labels, the segmentation module can help the deep appearance descriptor distinguish between objects with close appearances and similarities, even when the background is noisy.

Our proposed method performed better than YOLOv5 + DeepSORT. Especially under heavy fog conditions, our results show that the multi-object tracking accuracy (MOTA) increased by 35.15%, the multi-object tracking precision (MOTP) increased by 32.65%, the speed increased by 37.65%, and identity switches (IDS) were reduced by 46.81%. The remaining parts of this paper are structured as follows: We discuss the related studies in Section 2, we describe our materials and methodology in Section 3, Section 4 is where we present our improved appearance feature extraction network, we present our results and discussion in Section 5, and Section 6 consists of the conclusions.

## 2. Related Works

### 2.1. Object Detection

In the literature, many deep learning models [33,34,35,36] have provided excellent detection accuracy and speed for object detection tasks under favorable weather circumstances. AlexNet, as suggested by Krizhvsky et al. [37], was the first convolutional network used for image feature extraction, ushering in the current era of deep feature extraction. In our previous work [31], we completed a comprehensive review of camera-only as well as camera-and radar-fusion-based object detection methods. Some of the camera-only approaches include SSD proposed by Liu et al. [38], YOLO proposed by Redmon et al. [39], and its derivatives [40,41,42,43,44], RCNN proposed by Girshick et al. [45], and its derivatives [46,47,48]. Some camera-radar approaches include [28,49,50,51,52]. Although these methods provide excellent detection accuracy and speed in favorable weather, they are extremely inefficient when used in foggy weather [25]. There is a limited camera-radar approach in the literature for object detection under foggy weather conditions. Due to of the tradeoff between detection speed and accuracy, existing methods have a very limited range of use in foggy weather conditions. Recently, we proposed a deep learning-based radar and camera fusion (CR-YOLOnet) [31] based on YOLOv5 [44] for object detection in foggy weather conditions. In [31], we gave a comprehensive overview of YOLOv5. When detecting small and distant objects, our small CR-YOLOnet model achieved a balance between accuracy and speed and performed better than YOLOv5. The small CR-YOLOnet model reached an accuracy (mAP at 0.50) of 0.849 and a speed of 69 fps for medium and heavy fog conditions.

### 2.2. Tracking by Detection

Tracking by detection methods uses existing information about where objects are located along with the predicted information to produce a time-variant association model for object tracking. The multiple hypothesis tracking (MHT) approach is one of the first MOT techniques proposed in the literature [53,54,55]. Delaying complex data association determinations until more data is collected is crucial to the MHT approach. Many methods [9,10,56] have used the Kalman filter (KF), which considers both the current detections (bounding boxes) as input measurements and prior predictions to estimate where the target objects would appear in the next frame. Previous research has used the KF method as a velocity and motion model to enhance object associations. Particle filter algorithms were also studied for effective initialization and learning-phase multi-object tracking [57]. However, tracking multi-scale targets and identity switches (IDS) continues to be challenging. Thanks to the advances in computing power and the concept of deep learning, better tracking by detection methods (such as simple online and real-time tracking (SORT) [9], DeepSORT [10], etc.) have been developed.

Bewley et al. [9] suggested the SORT algorithm, which employs KF [58] and Hungarian algorithms [59] for MOT. The four main steps of SORT include (i) detection, (ii) estimation, (iii) data association, and (iv) creation and deletion of track identities. SORT employs KF to estimate object states based on the linear constant velocity model. The Hungarian algorithm helps to associate new detections (bounding boxes) with the KF predictions. However, the SORT algorithm is solely concerned with tracking speed and ignores target appearance. Owing to the fact that SORT considers position and IOU only, the SORT algorithm’s tracking performance degrades significantly when the tracked object overlaps in consecutive frames. Thus, SORT suffers frequent ID switches when tracked objects reappearence after an overlap and might even fail in the presence of occlusion. Wojke et al. [10] proposed that DeepSORT, an extension of the SORT algorithm, can track objects by associating their velocity and motion profile with their appearance features, which are extracted via a convolutional neural network. As a result, we chose DeepSORT as this paper’s baseline object tracking algorithm. We discuss DeepSORT further in Section 3.4.

Other tracking by-detection methods include Chen et al. [60], who suggested the MOTDT, which employs a scoring mechanism based entirely on convolutional neural networks to choose candidates optimally. Euclidean distances between the retrieved object appearance features were applied to enhance the association phase further. He et al. [61] suggested the GMT-CT, which leverages graph partitioning and deep feature learning to improve the association phase’s ability to represent the correlation between measurements and tracks.

Similarly, the use of Siamese networks trained using deep learning has increased in object tracking [14]. Lee et al. [15] developed FPNS-MOT that combines a Siamese network structure with a feature pyramid network. It compares the attributes of two inputs to generate a similarity vector. Tracks are updated in FPNS-MOT by selecting the highest-scoring combination of tracks and observations. Jin et al. [16] employed a Siamese structure to improve the efficiency of the Deep-SORT object feature extractor. To enhance the accuracy of the object association, Jin et al. extended their study by incorporating optical flow into the motion module [62].

### 2.3. Issues Related to the Use of Semantic Segmentation Cameras and Alternative Solutions

Semantic segmentation cameras, although not available in real-world applications, offer advantages in simulation environments. They provide rich annotated data [32] for training machine learning models and simplify ground truth generation [63]. Simulating semantic segmentation cameras in environments such as CARLA offers cost-effective development [32], safety in testing [63], scalability and flexibility, data generation for machine learning [64], algorithm validation and benchmarking [65], and cross-domain applications in manufacturing, aviation, and agriculture. Simulations allow for extensive testing and validation without expensive hardware, ensure safety during development, and provide a standardized platform for comparing different approaches. 

One issue with semantic segmentation cameras is that they do not exist in practical real-world applications. While they are useful for generating annotated datasets for machine learning models, conventional sensors used, such as RGB cameras, LIDAR, and RADAR, do not automatically provide detailed semantic labels for each pixel. This creates a simulation-to-reality gap [66], where models trained using data from semantic segmentation cameras may perform well in simulated environments but struggle in real-world situations due to disparities in data distribution. Additionally, the effectiveness of these models is highly dependent on the precision of the segmentation. Inaccuracies in segmentation can lead to problems in subsequent tasks such as object detection and tracking. Furthermore, achieving real-time semantic segmentation with the same level of precision as synthetic cameras requires significant processing resources and the use of advanced hardware and optimized algorithms [67].

To address the issues related to the use of semantic segmentation cameras, several alternative solutions are proposed for real-world applications. One alternative solution is the use of LIDAR and RGB cameras. By fusing LIDAR sensors with RGB cameras, it is possible to approximate the information obtained by semantic segmentation cameras. LIDAR sensors provide precise depth data, which can be combined with RGB photos to generate detailed 3D maps of the environment. This fusion process involves obtaining in-depth information [68]. LIDAR sensors and semantic segmentation [67] from RGB images, and then integrating the data to create a comprehensive depiction of the surroundings. Sensor fusion techniques, such as Kalman filters or Particle filters, can be employed to enhance the precision and resilience of the sensory system [69].

Another alternative solution is deep learning-based segmentation. Advancements in deep learning have made it possible to achieve real-time semantic segmentation using RGB cameras. Models such as DeepLabV3+ [67] utilize encoder-decoder architectures and atrous convolutions to effectively collect contextual information at multiple scales, resulting in improved segmentation accuracy. These models can be trained using extensive datasets and can be implemented on edge devices with optimal configurations. Overall, these alternative solutions offer potential ways to address the issues related to the use of semantic segmentation cameras, providing more accurate and comprehensive information for real-world applications.

LIDAR technology faces significant limitations in foggy and wet conditions [70]. Fog and rain cause attenuation and scattering of LIDAR signals [22,71], reducing range and accuracy. Wet surfaces can cause multiple reflections, leading to inaccurate readings in LIDAR systems [72]. The presence of water droplets increases noise in the LIDAR data, making it difficult to distinguish between actual obstacles and false positives [73]. Additionally, LIDAR’s ability to penetrate fog and heavy rain is limited compared to radar, which can better handle such adverse weather conditions [74].

## 3. Methods

### 3.1. Experimental Platform

The PyTorch framework and third-party library Opencv were used to experiment with Python programming. The hardware and software settings are as follows: Graphics card: Nvidia GeForce RTX 2070 with Max-Q Design; RAM: 16 gigabytes of memory; CPU: Intel Core 17-8570H 2.2 GHz 6 cores. 

### 3.2. Datasets and Semantic Labels

Due to of the unique nature of radar signals and the relative lack of publicly accessible datasets [75] that include both camera and radar datasets [21,76,77,78,79,80] under foggy weather conditions, the scope of AV research with respect to foggy weather conditions has been severely constrained. For autonomous vehicle research, there are only a very small number of datasets available [21,79] that combine information from cameras and radar when undertaken in foggy weather conditions.

In this study, we simulated an autonomous driving environment using the CARLA [32] simulator. The CARLA simulator enables autonomous driving simulation using a variety of sensors, including radar, RGB cameras, and semantic segmentation cameras. Semantic segmentation cameras can obtain data from an ego vehicle’s surroundings and analyze it over a wide range of light intensities. Semantic segmentation cameras can determine an object’s composition by analyzing its reflected light, since various materials absorb light at varying wavelengths. Figure 3 illustrates the semantic labels obtained from the semantic segmentation camera in the CARLA simulator.

Alongside the camera and radar sensors, we attached semantic segmentation cameras to the ego vehicle to obtain semantic labels of the objects in its environment. The semantic segmentation camera presents each object in its field of vision with a distinct color corresponding to the predetermined 11 object categories. However, we make use of the seven common traffic participant (bicycle, bus, car, motorcycle, person, traffic light, and truck) labels in our work.

### 3.3. Object Detection Model

In this work, we employed our recently proposed CR-YOLOnet [31], a deep learning-based camera-radar fusion network (CR-YOLOnet) for object detection with YOLOv5 as the baseline. Unlike the single-modal system used in the YOLOv5 baseline, the CR-YOLOnet takes input from both camera and radar sources. CR-YOLOnet uses two different CSPDarknet backbone networks to extract feature maps, one each for camera and radar. To further improve the network, the low-level feature information from the backbone network was sent to the feature fusion layers using two connections inspired by residual network concepts. These two connections aim to improve the network’s gradient backpropagation while simultaneously limiting the amount of feature information loss for relatively small or distant objects obscured by fog. To detect multi-scale item sizes in foggy weather circumstances, we improved CR-YOLOnet with an attention framework. Attention modules are added to the fusion layers to draw greater emphasis on and improve the feature representation of the features that aid in object detection. The attention module additionally addresses the issue of high-level feature information loss. We used a similar experimental platform as this work and obtained our camera and radar data from the CARLA simulator. For CR-YOLOnet training and testing, we used clear and foggy weather scenarios. With an accuracy (mAP at 0.50) of 0.849 and a speed of 69 fps, our small CR-YOLOnet model optimally maintained a balance between accuracy and speed.

### 3.4. The Object Tracking Method

The DeepSORT algorithm addresses the assignment problem of matching new detection measurements and predicted target states with the KF and Hungarian algorithms. The Hungarian algorithm with the Kalman filter algorithm provides a single-hypothesis tracking approach for moving targets. The Hungarian matching algorithm helps match up new detections with the predicted tracks. There were three distinct phases to the tracks. (i) tentative tracks: for each unmatched detection, a new set of track propositions is generated and stored in the tracked list for further observations. (ii) confirmed tracks: detections that are matched successfully are kept in the track matrix, (iii) deleted tracks: detections that cannot be matched or are no longer appearing within a specified number of frames are deleted from the track list. In addition to the data association of position and motion information, DeepSORT incorporates a CNN-based component that extracts, and associates object appearance features to address the ID switch problem. 

However, when it comes to tracking tasks in noisy situations such as fog, the network used for appearance feature extraction greatly influences the quality of the appearance feature information being extracted and, consequently, the tracking speed. The original feature extraction network is merely a very simple kind of convolution. As a result, targets with a high degree of appearance similarity on the detection frame are more likely to be mismatched with the wrong predictions. In addition, matching inaccuracies can occur when widely varied-size objects are on the frame. Thus, there is a need for further refinement when it comes to foggy weather applications.

In this work, our multi-object tracking technique is broken down into the following stages:(i)The KF takes detection information (the object bounding box provided by CR-YOLOnet) as the input measurement, then predicts the target object’s future states (position) in the detection frame, and the prior estimate value of the target object is estimated. Also, CR-YOLOnet extracts and saves the feature information of the target object on the detection frame.(ii)The Hungarian algorithms employ the appearance feature and Mahalanobis distance to associate the target object with the detection frame and track. In the event that the association procedure generates a successful match, the system will proceed to a Kalman update and provide tracking results. However, if there is no match, cascade matching is employed to associate the unmatched detection frame and track. Each track has its own time update parameter, “time_since_update”, established throughout the cascade matching process. Since the tracks are sorted by “*time_since_update*”, the unmatched detection frame is first associated with the track with the minimum “*time_since_update*” to prevent tracking failure while decreasing the frequency of ID switches.(iii)To evaluate if the target object in the detection frame and the track have the same ID and are a match, we compute the percentage of overlapping areas to compute their similarity. If the similarity computation generates a successful match, the system will proceed to a Kalman update and provide tracking results. However, if there is no match for the subsequent five frames, the mismatched track will be associated with a new detection frame. The tracking result will be returned if the detection frame is found within the next five frames. The target is consequently deleted if no match is found after five frames.

To acquire the overall tracking result, it is necessary to modify and adapt the tracking trajectory obtained from the KF by repeating the three stages mentioned above. Allocating a new identifying ID or removing a set trajectory are the actions that must be taken once a detection frame is unable to match and track.

#### 3.4.1. Kalman Filter Prediction of Target Object State

We employ a Kalman uniform velocity model for the object motion state estimation. Equations (1) and (2) describe the discrete-state space model. For the motion state of each target $x_{k}$, the target object state is set from the previous state k−1 to the present state k, the measurement vector $z_{k}$ is the target object detection result that consists of the bounding box coordinates (*a*, *b*) at the current scan k, and the process noise at scan k.
(1)$$x_{k}=Ax_{k-1}+\omega_{k-1}$$
(2)$$z_{k}=Hx_{k}+v_{k}$$
where A is the state transition matrix, $\omega_{k}\;\sim \;N\left(0,\;Q_{k}\right)$ denotes the state white Gaussian process noise and $Q_{k}$ is the estimated process error covariance matrix, H is the measurement matrix, $v_{k}\;\sim \;N(0,\;R_{k})$ denotes the measurement of white Gaussian process noise,$\;R_{k}$ is the estimated measurement error covariance matrix, with an initial state $x_{0}∼\;N\;\left(x_{0}^{-},\;P_{0}\right)$, $x_{0}^{-}\in \;R^{n}$ is the prior mean, and $P_{0}\in \;R^{n\times n}$ is the initial covariance.

We represent the state of the target object by vector x, and it can be expressed as follows:(3)$$x={\left[a,b,\gamma ,h,\dot{a},\dot{b},\dot{\gamma },\dot{h}\right]}^{T}$$
where (a,b) is the bounding box coordinate of the target, γ is the bounding box aspect ratio, h is the bounding box height, $\dot{a},\dot{b},\dot{\gamma },\dot{h}$ represents the corresponding speed information of a,b,γ,h.

**Tracking Space**: The tracking space in autonomous vehicle systems is the coordinate system used to track objects. Our method performs tracking using image plane coordinates, which offers several advantages and limitations. Image coordinate tracking entails observing the variations in the location and dimensions of objects as they are depicted in the two-dimensional projection of the scene recorded by the camera on the vehicle. This approach is very advantageous for identifying and monitoring objects in real-time because of its efficient processing capabilities and compatibility with a wide range of vision-based algorithms [3,81]. Nevertheless, to track objects accurately, it is important to comprehend the correlation between the coordinates in the image plane and the corresponding spatial coordinates in the real world. The main obstacle in this situation is that the two-dimensional image plane does not immediately provide depth information, therefore complicating the extraction of precise distances and relative velocities of objects. These measurements are essential for the navigation and decision-making processes of autonomous vehicles [82].**Physical Interpretation of Tracking Parameters**: Image-based tracking involves representing the physical parameters of tracked objects in the image plane, such as the object’s position, height, and aspect ratio, as described in Equation (3). The parameters change over time because of the relative motion between the camera (which is attached to the vehicle) and the objects in the scene. The key parameters and their physical interpretations are as follows:**Position**: The (a, b) coordinates of the object in the image plane. These coordinates represent the location of the object’s bounding box within the 2D image frame [83].**Aspect Ratio**: The aspect ratio (γ) is the ratio of the width to the height of the object’s bounding box. Changes in aspect ratio can indicate changes in the object’s orientation relative to the camera. For example, a vehicle turning relative to the camera may show a different aspect ratio over time [84].**Height**: The height (h) is the vertical dimension of the object’s bounding box in the image plane. This parameter changes as the object moves closer to or further away from the camera, affecting its apparent size. Changes in the height of the bounding box can indicate relative changes in distance between the object and the camera [85].

The changes in the position parameters over time can be described by their respective velocities:(4)$$v_{a}=\frac{da}{dt}$$
(5)$$v_{b}=\frac{db}{dt}$$

As mentioned earlier, the representation of objects in image coordinates undergoes changes over time as a result of relative motion and changes in perspective, requiring the incorporation of velocity modeling for height and aspect ratio. The relationship between the speed of height $\left(v_{h}\right)$ and aspect ratio $v_{\gamma }$ may be comprehended by examining the changes in the size of the object’s bounding box in the image plane. $v_{h}$ represents the rate of change in height with respect to time $\frac{dh}{dt}$ and it indicates the vertical height of the enclosing box. $v_{ar}$ represents the rate of change of aspect ratio with respect to time $\frac{d\gamma }{dt}$. The observed changes are a result of the relative displacement between the camera and the object, particularly noticeable in situations involving rapid motion or abrupt changes in direction. The tracking algorithm calculates these modifications to precisely predict the future position of the item. The relationship between the changes in height and aspect ratio may be expressed as follows:(6)$$v_{h}=\frac{dh}{dt}$$
(7)$$v_{ar}=\frac{dar}{dt}$$
where $v_{x}$ and $v_{y}$ are the velocities in the x and y directions, $v_{h}$ is the velocity of the height, and $v_{ar}$ is the velocity of the aspect ratio. These velocities are crucial for predicting the future positions of objects in the image plane.

There are two primary parts to the Kalman filter. First, we make predictions of the system state using the time-update mathematical state model. Equations (8) and (9) describe the prediction of the future state of the target object and the future error covariance:(8)$${\hat{x}}_{k}^{-}=A{\hat{x}}_{k-1}^{-}+\omega$$
(9)$$P_{k}^{-}=AP_{k-1}^{-}A^{T}+Q$$
where ${\hat{x}}_{k}^{-}$ is the prior estimate that describes the future state of the target object at scan k, $P_{k}^{-}$ is the prior estimate of the future error covariance at scan k, ${\hat{x}}_{k-1}^{-}$ is the posterior estimate of the target object state at scan k−1, $P_{k-1}^{-}$ is the posterior estimate of error covariance at scan k−1. 

Second, the predicted state values are compared to the measured state values to generate state estimation output. In Equations (10)–(12), we compute the Kalman gain, update the estimate using measurements $z_{k}$, and update the error covariance, respectively.
(10)$$K_{k}=P_{k}^{-}H^{T}{\left(HP_{k}^{-}H^{T}+R\right)}^{-1}$$
(11)$${\hat{x}}_{k}={\hat{x}}_{k}^{-}+K_{k}\left(z_{k}-H{\hat{x}}_{k}^{-}\right)$$
(12)$$P_{k}=\left(I-K_{k}H\right)P_{k}^{-}$$

A state estimation output is generated from the adjusted difference between the predicted and observed states, considering the predicted noise and error in the system and the measurements. The generated state estimation output is fed into the mathematical state model described in Equations (4) and (5) to predict the target object’s future state at the subsequent time update. Thus, the cycle starts all over again. The error between the real value and the observed value is minimized via the iterative process, bringing the predicted value closer and closer to the real value until the optimal tracking outcome is attained.

Autonomous vehicle development faces challenges due to the limitations of tracking objects in image coordinates. This can lead to the loss of critical information such as the distance and relative velocity of other traffic participants, which is crucial for safe navigation and decision-making. The projection of RADAR data into image coordinates can lead to the loss of this information, as these measurements are not directly represented in the 2D image plane. This can result in a significant reduction in the quality and usability of data for subsequent algorithms [86,87].

The loss of distance and relative velocity information poses several challenges for planning and control algorithms. Without accurate distance measurements, the vehicle may misjudge the proximity of obstacles, leading to potential collisions. Inefficient path planning can also occur due to the lack of relative velocity data, impairing the vehicle’s ability to predict the future positions of moving objects and increasing the risk of accidents. Furthermore, the performance of planning and control algorithms may degrade, reducing the efficiency and safety of the autonomous vehicle. Potential solutions to address this issue include preserving and effectively utilizing the necessary information [87,88].

To prevent the loss of critical information in autonomous vehicles, several solutions can be implemented. Inverse Perspective Mapping (IPM) can transform image coordinates back to real-world coordinates, reprojecting 2D image points into 3D space. Sensor Fusion Techniques can enhance object tracking by combining data from multiple sensors using techniques such as Kalman Filters and Particle Filters. Advanced depth estimation algorithms, using stereovision or monocular depth estimation, can provide 3D spatial information from 2D images using deep learning techniques. These solutions can help retain essential distance and velocity data for effective planning and control in autonomous vehicles, improving their safety and operational efficiency [86,87].

#### 3.4.2. Matching New Detection Measurements and Predicted Target States

The motion and appearance information matching are incorporated using (i) Mahalanobis distance and (ii) minimum cosine distance, respectively. Let p and q be the order number of the predicted target state and target bounding box detection. 

(i)Mahalanobis distance:

The Mahalanobis distance ($dist_{M}$) includes the motion data by estimating the distance between the new target detections and predicted target states, and it can be calculated as follows: (13)$$dist_{M}(p,\;q)={\left(d_{q}-y_{p}\right)}^{T}S_{p}^{-1}(d_{q}-y_{p})$$
where $d_{q}$ represents the q−th bounding box detection, $\left({y_{p},S}_{p}\right)$ denotes the p−th track in measurement space, $y_{p}$ denotes the projection of the predicted value of the p−th track in the detection space, $S_{p}$ denotes the covariance matrix of the p−th track in the measurement space. 

With the KF’s uncertainty estimation of the target state, the Mahalanobis distance computes the distance from the mean track to the detection’s standard deviation. Then a threshold $t^{\left(1\right)}$ described in Equation (2) is used to determine if the p−th track and q−th detection are related or not.
(14)$$b_{M}(p,\;q)=\left\{\begin{array}{c} 1,\;dist_{M}(p,\;q)\;\leq t^{\left(1\right)}\\ 0,\;dist_{M}(p,\;q)>\;t^{\left(1\right)} \end{array}\right.$$ If the p−th track and q−th detection is related, the threshold evaluates to 1. Otherwise, it is 0.

(ii)Appearance feature matching:

The appearance features of the target are disregarded while using Mahalanobis distance since it only considers the distance relationship between the detected target and the predicted target states. The appearance features are extracted using a simple convolutional neural network to incorporate the appearance metric. A total of two convolutional layers and six residual blocks makes up this network. Appearance feature descriptors $r_{q}$ are extracted from each bounding box detection $d_{q}$ using a simple convolutional neural network shown below. For each track k, all the matched appearance descriptors are stored in $R_{p}$. Hence, the minimum cosine appearance $(dist_{C}$) distance between the p−th track and q−th detection can be calculated using Equation (3).
(15)$$dist_{C}(p,\;q)=min\left\{1-r_{q}^{T}r_{k}^{\left(p\right)}|r_{k}^{\left(q\right)}\;\in R_{p}\right\}$$ Using the threshold $t^{\left(2\right)}$ in Equation (16), we can show whether p−th track and q−th detection in Equation (15) are related. Similarly, if the p−th track and q−th detection are related, the threshold evaluates 1. Otherwise, it is 0.
(16)$$b_{C}(p,\;q)=\left\{\begin{array}{c} 1,\;dist_{C}(p,\;q)\;\leq t^{\left(2\right)}\\ 0,\;dist_{C}(p,\;q)>\;t^{\left(2\right)} \end{array}\right.$$ When the Mahalanobis distance is used in conjunction with the minimum value of the cosine distance, the DeepSORT algorithm’s efficiency can be enhanced. The Mahalanobis distance includes details about object positions depending on the motion to address short-term prediction and matching. When the motion information is less reliable because of extended occlusions, the cosine distance considers appearance information, which is important for re-establishing identities. Therefore, the fusion of Mahalanobis distance $dist_{M}(p,\;q)$ and cosine distance $dist_{C}(p,\;q)$ from Equations (13) and (15), respectively, is described using the weighted sum $U_{p,q}$ in Equation (17): (17)$$U_{p,q}=\eta dist_{M}(p,\;q)+(1-\eta )dist_{C}(p,\;q)$$
where η which is often set to 0.1 denotes the hyperparameter used for setting the weights of the Mahalanobis and cosine distances. In Equation (18), a gated matrix helps to establish whether the association of metrics is related.
(18)$$b(p,\;q)=\prod_{n=1}^{2}b^{\left(n\right)}(p,\;q)$$

In addition, the cascade-matching method is used to compare the predicted target’s motion trajectory with KF and new target detection [89]. A new measuring matrix is constructed using both target appearance features and velocity information to evaluate the degree of similarity between a detection and a trajectory. Although, when compared to the SORT algorithm, the DeepSORT algorithm performs significantly better.

### 3.5. The Appearance Feature Extraction Model

As mentioned earlier, the appearance feature extraction model in the original DeepSORT employed merely convolution and pooling layer procedures. The feature map generated by the backbone network’s output is prone to losing relevant details about the target object. For very small and distant targets, this leads to an incorrect knowledge of object appearance features. Multi-object tracking models notably need faster tracking speeds. In addition to tracking small and distant objects at a fast speed, extracting quality appearance features will enhance the DeepSORT algorithm’s ability to distinguish between objects with similar appearances and track them accurately. We proposed an appearance feature network to replace the basic convolutional neural network used for the appearance descriptor in the DeepSORT algorithm. Our deep appearance descriptor employs a CSPNet-based backbone for low-level feature extraction and an FPN-based neck for multi-scale appearance feature fusion to address objects of varying sizes.

The cross-stage partial connection (CSP) is a method that was initially derived from CSPNet [90] and is used to optimize complex computational processes. The CSP network can help increase feature-learning capacity during training. Figure 4a illustrates how a network can be “CSP-ize”. The base layer’s feature map is split into two components, the main component and a skip connection, combined by transition, concatenation, and transition to efficiently cut down on redundant gradient information. Owing to the fact that CSP-ization shortens gradient flow, therefore, it increases accuracy and decreases inference time while making model scaling possible [11]. As a result of scaling the model, the ability to detect objects of smaller sizes is made possible. Given the larger size of the input network, the wider receptive field achieved by the CSP connection directly results from the higher number of convolutional layers. The feature size plays a significant role in the representation of the target’s feature information when performing feature extraction. 

When generating the final feature map, the Feature Pyramid Network (FPN) [91] shown in Figure 4b aggregates features from several depths and layers. The final feature map includes a range of multi-layer semantic information due to the feature fusion that occurred at various levels. The feature pyramid network gets its name from its structure and shape, which are both reminiscent of pyramids. In FPN, the backbone networks are responsible for feature extraction, and the top-down fusion of feature maps is utilized to combine the resulting features $C_{0},\;{\;C}_{1},$ and $C_{2}$. Network structures of varying depths have varying degrees of accuracy when used to extract feature information from targets. Scale-dependent disparities in feature information during the matching phase can be mitigated using fused object appearance feature information retrieved from several network depths. The FPN integrates a shallow feature extraction network for extracting spatial information with a deep feature extraction network for obtaining appearance feature information.

We adopted GhostNet, discussed in Section 3.5.1, into our deep appearance descriptor to replace the traditional convolutional layers. We adopted the segmentation module discussed in Section 3.5.2 to provide rich semantic information to the low-level appearance feature map using semantic labels.

#### 3.5.1. GhostNet for Improved Performance and Reduced Computational Complexity and Cost

The Ghost module was developed to take the place of the traditional convolutional layers in standard neural networks [92]. The aim of the Ghost module, which is illustrated in Figure 5, is: (i) to improve the performance of neural networks by generating more features, thus improving the integrity of the feature extracted; (ii) to utilize a lesser number of parameters, thus reducing computational complexities and cost without diminishing the output feature map. In the Ghost module, the conventional convolution process is divided into two separate steps. In the first step of the process, a conventional 1 × 1 convolution is performed on the input to acquire the required feature concentration. The second step involves performing a series of simple linear operations, such as layer-by-layer convolution, on the intrinsically concentrated feature maps obtained from the prior step to produce additional feature maps. 

Consider an input feature map $X\in \;R^{c\times h\times w}$ with c number of channels, height denoted by h and weight denoted by w, the procedure for generating n feature maps using conventional convolution can be expressed as:(19)Y=X∗f+b
where f denotes the convolution kernels with size, b denotes the bias term. ∗ is the convolution operation. Using n convolution filter $f\in \;R^{c\times k\times k\times n}$ with k · k kernel size, the output feature map is $Y\in \;R^{h^{\prime }\times w^{\prime }\times n}$. The heights and widths of the output feature maps are denoted by $h^{\prime }$ and $w^{\prime }$, respectively. 

Since the number of filters and channels often needs to be quite high, the needed number of floating-number operations (FLOPs) may easily reach hundreds of thousands. The needed number of FLOPs for the conventional convolution process can be expressed as follows:(20)$$FLOPs=n\;\cdot \;h^{\prime }\cdot w^{\prime }\cdot c\cdot k\cdot k$$ Assumptions can be made that the generated feature maps are “ghosts” of certain original feature maps that have been reshaped in a computationally cost-effective way. These assumptions can be made to prevent redundancy and similarities in the individual output feature maps generated by ordinary convolutional layers while exhausting a vast number of FLOPs and parameters. Equation (21) describes the ordinary convolution for creating the m intrinsic feature maps $Y^{\prime }\in \;R^{h^{\prime }\times w^{\prime }\times m}$ such that m ≤n and $f^{\prime }$ are the m convolution kernels of the size of k·k.
(21)$$Y^{\prime }=X\;{*\;f}^{\prime }$$ Applying a sequence of inexpensive linear operations to each intrinsic feature in $Y^{\prime }$ yields s ghost features, which may then be used to construct the necessary n feature maps as described in Equation (22) and needed FLOPs in Equation (23):(22)$$y_{ij}=\Phi_{i,\;j}\left(y_{i}^{\prime }\right),\;\;for\;all\;\;i\;\in \left(1,\;m\right),\;j\in \;\left[1,\;s\right]$$
(23)$$FLOPs=m\;\cdot \;\;h^{\prime }\cdot \;w^{\prime }\cdot \;c\cdot \;k\cdot \;k+\left(s-1\right)\cdot m\cdot \;h^{\prime }\cdot \;w^{\prime }\cdot \;c\cdot \;d\cdot \;d$$
where $Y_{i,\;j}$ denotes the j−th ghost feature map generated by the convolution kernel size d·d of each linear operation $\Phi_{i,\;j}$ excluding the last operation $\Phi_{i,\;s}$ used retaining the maps of intrinsic features, $y_{i}^{\prime }$ is the i−th intrinsic feature map in $Y^{\prime }$. 

Figure 6a illustrates our GhostNet structure. The CBL module is made up of convolution, batch normalization, and the leaky ReLU activation function sub-modules. The Ghost module uses standard convolution to produce a portion of the original feature map. Next, it convolves each of these feature maps individually to get a portion of the associated feature map. Furthermore, then it adds the latter feature map to the first feature map. Our improved convolution operation, called Ghost convolution, consists of the CBL block, CSP block, and GhostNet block, as shown in Figure 6b. 

#### 3.5.2. Segmentation Module for an Improved Appearance Feature

We adopted the segmentation module (SM), which is primarly composed of atrous convolutional layers, to add rich semantic information to the low-level appearance feature map generated using semantic labels from [93]. The purpose of the semantic labels is to add their own robust, meaningful semantic features to the low-level feature map extracted from the backbone network, as shown in Figure 7. The goal of incorporating the segmentation module into the appearance feature extractor is to improve the integrity of appearance features and to be able to distinguish between objects of similar appearance in a noisy detection frame. 

Thus mitigating the problem of mismatch between detection measurements and Kalman filter predictions under foggy weather conditions. 

Several parameters make up the segmentation module. First, we consider a primary low-level input feature map $X\in \;R^{C\times H\times W}$ with C number of channels, height denoted by H and weight denoted by W, and a semantic label (ground-truth) $G\in \;{\left\{0,\;1,\;2,\;\cdots ,\;N\right\}}^{H\times W}$ such that N is the number of object classes in the label (in our case, we make use of 7 object classes). In Equation (24), the intermediate feature map $G\left(X\right)\in \;R^{C^{\prime }\times H\times W}$ is used to estimate the per-pixel segmentation prediction $Y\;\in \;R^{(N+1)\times H\times W}$, this is also known as F path. In addition, for the H path, the intermediate feature map $G\left(X\right)$ is employed to create a profound feature map $Z\in \;R^{C\times H\times W}$ with semantic content as described in Equation (25):(24)Y=F(G(X))
(25)Z=H(G(X)) The element-wise multiplication of the primary low-level appearance feature map X and the semantically profound feature map Z activates X to give $X^{\prime }$. The activation process produces a map of low-level appearance features $X^{\prime }=X\;⨂\;Z,$ which is a semantically activated appearance feature map. $X^{\prime }$ does provide not only rudimentary visual patterns but also high-level semantic meaning. The cross-entropy loss function $L_{seg}\left(I,\;G\right)$ of the segmentation module is given as:(26)$$L_{seg}\left(I,\;G\right)=-\frac{1}{HW}\sum_{h,\;w}\log\left(Y_{G_{h,\;w},\;h,w\;}\right)$$
where I is the image, G is the semantic label, and Y is the segmentation prediction.

## 4. Improved Deep Appearance Feature Extraction Network

### 4.1. The Architecture of Our Appearance Feature Extraction Network

As previously mentioned, we chose the CSPDarknet-based backbone with the intention of enhancing the functionality of the appearance feature extraction network illustrated in Figure 8. The ghost convolution block consists of the CBL block, CSP block, and GhostNet block (see Section 3.5.1). The goal of the CSP block is to help improve the capacity of our networks to learn as many features as possible from an image during training. Introducing the GhostNet block helps to generate more features to enhance the integrity of the feature extracted while utilizing a lesser number of parameters to alleviate computational complexities and cost. To increase tracking accuracy and manage the autonomous driving task in dynamic and foggy weather environments, we integrate and fuse appearance features from several layers, resulting in a richer appearance feature vector.

Using the segmentation module from Section 3.5.2, rich semantic information from semantic labels can help to improve the appearance feature vector generated by the appearance feature extraction network. In this work, we integrate the segmentation module into the backbone of the CSPDarknet-based backbone and the FPN-based neck because it is designed to generate multi-scale level feature maps, including small, medium, and large objects. In addition to the semantic label input, the segmentation module uses low-level appearance feature maps (denoted by the dark red square dot arrow in Figure 8) generated in the backbone to understand semantic segmentation based on the influence of segmentation ground truth. Thus, the segmentation module uses its own rich segmentation features to bolster the low-level features to produce low-level semantically activated appearance feature maps (denoted by the dark green long dash arrow in Figure 8). The low-level semantically activated appearance feature maps have the capability of acquiring not just the fundamental visual pattern of a target object, but accurate semantic information associated with it.

We implemented spatial pyramid pooling (SPP) [94] to (i) enhance the receptive field of our network, (ii) decouple the context features, and (iii) make it easier for the neck network to fuse appearance features from several layers. We introduced SPP at the beginning of the backbone network to prevent loss of resolution and noise in the input image that can occur if scaled and cropped. The SPP at the very end of the backbone network consists of three different pooling layers with sizes of 5 × 5, 7 × 7, and 13 × 13.

To generate the many local features, SPP combines the results of the three pooling layers and feeds them as input to the subsequent convolutional module, where further feature learning is carried out. 

$C_{0}$, $C_{1}$, and $C_{2}$ are the extracted appearance feature maps from three different depths were fed into the FPN and fused. The appearance features generated by the $C_{0}$ layer possess rich and high-level semantic information that enhances the extraction of features from large target objects. The feature maps from the $C_{0}$ layer are inadequate for feature extraction from small target objects. Despite the fact that $C_{1}$ and $C_{2}$ outputs may not have as much detail in their feature maps as $C_{0}$, they are excellent at extracting features from smaller target objects while still providing significant and useful positional information. The FPN network combines feature maps $C_{0}$, $C_{1}$, and $C_{2}$ from the backbone network’s output at varying depths, as shown in Figure 8. After performing complete joining and batch normalizing, the resulting object appearance feature vector is acquired. The object appearance feature vector is used to determine an estimation of the extent to which the track’s appearance is similar to the detection appearance. We performed the full cascading matching procedure using the motion information and the cost matrix, which is the estimation of appearance feature similarity between detection and tracks.

### 4.2. Training

Figure 9 illustrates samples of our CARLA dataset used for training our appearance feature extraction network in this work. The dataset includes both sunny and foggy conditions, with RGB camera data in the top row and the corresponding semantic segmentation camera data in the second row, as shown in Figure 9. The RGB data set serves as the input into the appearance feature network, while semantic segmentation serves as the input (provides semantic labels) to the segmentation module. We make use of the seven common traffic participant (bicycle, bus, car, motorcycle, person, traffic light, and truck) labels in our work. There are a total of 18,628 RGB pictures, with each having a corresponding semantic segmentation image. The training set consists of 80 percent of the images, and the remaining 20 percent is used for testing. The appearance feature extraction network was trained using both clear and foggy image datasets for 100 epochs with a batch size of 64. To predict the spatial location of the tracked object, we used the CIoU loss function [95] for bounding box regression described in our previous work [25]. 

Figure 10 shows the training loss curve (red), which is the training phase, and the prediction loss curve (blue), which is the prediction phase of the appearance feature extraction network using CIoU loss with the segmentation module incorporated. In both the training and prediction phases, the loss curves decreased rapidly during the initial 15 epochs. Thus, the rate of loss begins to slow down in an unstable manner because of insufficient model accuracy at the start of the training phase. However, at the 60th epoch, both loss curves begin to flatten out and become stable until the 100th epoch.

## 5. Multi-Object Tracking Experimental Results and Discussion

### 5.1. Comparison of Multi-Object Tracking Performance Using Our CARLA Dataset

Following the training phase of our appearance feature extraction network, the results of the training are then incorporated into our improved DeepSORT tracking model to evaluate the performance of our improved multi-target tracking algorithm. Throughout this Section 5.1, we referred to our improved multi-target tracking algorithm using the GIoU loss function [96] with the segmentation module as “Ours(GIoU) with SM” and without the segmentation module as “Ours(GIoU) without SM”. Similarly, we referred to our improved multi-target tracking algorithm using the CIoU loss function [95] with the segmentation module as “Ours(CIoU) with SM” and without the segmentation module as “Ours(CIoU) without SM”. The following metrics serve as the basis for the evaluation [10]:The multi-object tracking accuracy (MOTA) describes the total tracking accuracy with respect to false positives (FP), false negatives (N), and identity switches (IDS), and it is expressed in Equation (27).The multi-object tracking precision (MOTP) describes the total tracking precision measured with respect to the amount of actual bounding box overlap with the predicted position, and it is expressed in Equation (28).
(27)$$MOTA=1-\frac{\sum_{t}\left({FP}_{t}+{FN}_{t}+{IDS}_{t}\right)}{\sum_{t}G_{T}}$$
(28)$$MOTP=\frac{\sum_{t,i}d_{t,i}}{\sum_{t}c_{t}}$$
where $c_{t}$ denotes the total number of matches at frame t, $d_{t,i}$ is the distance between the predicted and the ground-truth bounding box, $G_{T}$ is the number of tracking targets. In addition, we performed evaluations using other metrics, including mostly tracked (MT), mostly lost (ML).  MT is used to describe the percentage of ground-truth tracks that do not switch labels for the majority (80%) of their existence. ML is used to describe the percentage of ground-truth tracks maintained for no more than 20% of their existence. 

Table 1, Table 2 and Table 3 describe the multi-object tracking performance of Ours (CIoU) with and without SM and Ours (GIoU) with and without SM in clear-day, medium fog and heavy fog conditions, respectively. Under clear day weather conditions in Table 1, Ours (CIoU) with SM has a MOTA of 74.86, a MOTP of 84.21, a MT of 43.40%, and a speed (FPS) of 68.34, which are higher than the other three models. In addition, Ours (CIoU) with SM has ML of 15.86% and IDS of 513, which is lower than the other three models. When compared to Ours (CIoU) without SM, the MOTA in Ours (CIoU) with SM increased by 4.30%, MOTP increased by 5.13%, MT increased by 7.83%, and speed (FPS) increased by 8.58%. The ML and IDS were reduced by 12.16% and 14.25%, respectively.

When operating in medium fog conditions in Table 2, Ours (CIoU) with SM has MOTA of 68.25, MOTP of 79.65, MT of 38.77%, and a speed (FPS) of 66.15, which are higher than the other three models. Moreover, Ours (CIoU) with SM has ML of 17.05% and IDS of 691, which is lower than the other three models. However, when compared to Ours (CIoU) without SM, the MOTA in Ours (CIoU) with SM increased by 5.10%, MOTP increased by 8.55%, MT increased by 14.38%, and speed (FPS) increased by 11.01%. The ML and IDS were reduced by 15.31% and 17.01%, respectively.

In Table 3, the heavy fog situation shows that Ours (CIoU) with SM has a MOTA of 66.14, a MOTP of 75.78, MT of 36.80%, and a speed (FPS) of 64.88, all of which are greater than the other three models. In addition, Ours (CIoU) with SM has ML of 19.24% and IDS of 816, which is significantly lower than the other three models’ respective values. When compared to Ours (CIoU) without SM, the MOTA in Ours (CIoU) with SM increased by 9.02%, a MOTP increased by 7.43%, the MT increased by 18.05%, and the speed (FPS) increased by 15.32%. The ML and IDS were reduced by 15.57% and 21.09%, respectively.

In Table 4, under clear day weather conditions, when compared to CR-YOLO + DeepSORT, the MOTA in Ours (CIoU) with SM increased by 8.58%, MOTP increased by 6.84%, MT increased by 13.85%, and a speed (FPS) increased by 11.59%. However, the ML and IDS were reduced by 22.58% and 37.99%, respectively. Compared to YOLOv5 + DeepSORT, the MOTA in Ours (CIoU) with SM increased by 18.12%, MOTP increased by 17.93%, MT increased by 37.19%, and speed increased by 32.71%. However, the ML and IDS were reduced by 43.26% and 38.44%, respectively.

Under medium fog conditions in Table 5, compared to CR-YOLO + DeepSORT, the MOTA in Ours (CIoU) with SM increased by 10.66%, the MOTP increased by 10.87%, the MT increased by 16.38%, and the speed increased by 16.45%. Nonetheless, the ML and IDS in Ours (CIoU) with SM decreased by 24.60% and 38.06%, respectively. Compared to YOLOv5 + DeepSORT, the MOTA in Ours (CIoU) with SM increased by 24.13%, MOTP increased by 24.71%, MT increased by 43.27%, and the speed increased by 35.80. However, the ML and IDS decreased by 35.12% and 44.0%, respectively.

Under heavy fog conditions, Table 6 shows that the MOTA in Ours (CIoU) with SM increased by 16.17%, the MOTP increased by 17.99%, the MT increased by 23.27%, and the speed increased by 23.39% in comparison to CR-YOLO + DeepSORT. However, the ML and IDS decreased by 27.54% and 40.78%, respectively. Compared to YOLOv5 + DeepSORT, the MOTA in Ours (CIoU) with SM increased by 35.15%, the MOTP increased by 32.65%, the MT saw an increase of 48.72%, and the speed saw an increase of 37.65%. However, the ML and IDS decreased by 41.65% and 46.81%, respectively, when compared to YOLOV5 + DeepSORT. This implies that employing the segmentation module and CIoU loss function efficiently improves our proposed model’s object tracking capability in foggy and clear weather conditions.

### 5.2. Qualitative Results of Multi-Object Tracking Performance on Our CARLA Dataset

In this section, we present a comparison of the qualitative results of CR-YOLO + Ours (CIoU) with SM, CR-YOLO + Ours (GIoU) with SM, CR-YOLO + DeepSORT on our CARLA dataset. Throughout this Section 5.2, CR-YOLO + Ours (CIoU) with SM, CR-YOLO + Ours (GIoU) with SM, and CR-YOLO + DeepSORT are referenced as Ours (CIoU), Ours (GIoU), and CR-YOLO DeepSORT, respectively. 

As previously mentioned in Section 3.4, if the similarity computation between the target object in the detection frame and the track generates a successful match, the system will proceed to a Kalman update and provide tracking results. If no match is found after five frames, the target is marked as untrackable and consequently deleted. This implies that, even though the CR-YOLO algorithm is efficient in detecting both small and distant target objects, it is up to the tracking module to generate a successful match within the first five frames for tracking results to continue. We compare Ours (CIoU) displayed in row 1, Ours (GIoU) displayed in row 2, and CR-YOLO + DeepSORT displayed in row 3 under clear weather conditions (Figure 11), medium fog weather conditions (Figure 12), and heavy fog weather conditions (Figure 13). 

In Figure 11a, Figure 12a and Figure 13a, several distant and small objects were detected and tracked. However, these objects were tracked until they were closed and became medium size in Figure 11b, Figure 12b and Figure 13b and larger size in Figure 11c, Figure 12c and Figure 13c. Ours (CIoU) performed better than Ours (GIoU) and DeepSORT tracking modules, especially regarding distant and small objects. Ours (CIoU) successfully generated, maintained, and matched more tracks than Ours (GIoU) and DeepSORT. For instance, in Figure 11a, Ours (CIoU) confirmed and maintained four tracks compared to two tracks in Ours (GIoU) and one track in DeepSORT.

Similarly, in Figure 12a, Ours (CIoU) confirmed and maintained 5 tracks compared to 3 tracks in Ours (GIoU) and 2 tracks in DeepSORT. In Figure 13a, Ours (CIoU) confirmed and maintained 3 tracks compared to 2 tracks in Ours (GIoU) and 1 track in DeepSORT. Similarly, for medium-sized objects, in Figure 11b, Ours (CIoU) confirmed and maintained 6 tracks compared to 4 tracks in Ours (GIoU) and 3 tracks in DeepSORT. In Figure 12b, Ours (CIoU) maintained 5 tracks compared to 4 tracks in Ours (GIoU) and 3 tracks in DeepSORT.

Ours (CIoU) successfully handled MOT even when there was a variation of target sizes in the detection frame in all three weather scenarios. In Figure 11c, while tracking large objects, distant vehicles and traffic lights that appeared small were tracked simultaneously by Ours (CIoU) and Ours (GIoU), unlike the CR-YOLO DeepSORT that tracked the larger objects only. However, in Figure 13c, under heavy fog conditions, only Ours (CIoU) was able to track the distant and small-sized pedestrians, unlike Ours (GIoU) and CR-YOLO DeepSORT, which tracked the larger objects only. Obviously, the CR-YOLO DeepSORT performed better in clear weather conditions than in medium and heavy fog conditions. For instance, in Figure 13c, DeepSORT could not confirm and track the sport utility vehicle due to occlusion and atmospheric scattering. However, despite the problems of occlusion and atmospheric scattering, both Ours (CIoU) and Ours (GIoU) successfully maintained the object’s identity for a longer period of time.

In all three weather conditions, fusing the appearance feature map at three different depth levels and with the segmentation module gave Ours (CIoU) and Ours (GIoU) better performance leverage over the CR-YOLO DeepSORT. Ours (CIoU) performed better than Ours (GIoU) due to the CIoU loss function, which not only considers non-overlapping regions between the actual and ground-truth frames but also uses the weight function. The weight function is a trade-off parameter that gives the overlap region factor a higher priority for regression. CIoU also measures the consistency or similarity of the aspect ratio between the bounding boxes. Thus, the ability of the tracking model to efficiently generate and match tracks is essential for critical safety systems such as autonomous driving.

### 5.3. Addressing Fundamental Issues and the Real-Life Applicability of Our Proposed Method

In challenging weather circumstances such as fog, a significant challenge is the scarcity of real-world datasets that encompass RGB cameras, RADAR, and semantic segmentation for varying levels of visibility, as highlighted in our work. The absence of these datasets impedes the direct implementation and validation of our models in real-life situations. To address these concerns, we have thoroughly discussed the techniques of domain adaptation and transfer learning, which are important for bridging the gap between simulated and real-world data. Additionally, we outline a potential proof of concept that could be implemented to validate the real-world applicability of the proposed solutions.

Domain adaptation and transfer learning are essential methods for closing the gap between simulated and real-world data. These techniques allow models that have been trained using simulated data to effectively function in a different environment, such as real-world data, by addressing the differences between these two environments.

#### 5.3.1. Domain Adaptation

Domain adaptation is altering a model to enhance its ability to generalize well across diverse data distributions. It is crucial when there is a transition from the training (source) domain to the deployment (target) domain. The objective is to reduce the gap between the source and target domains in order to guarantee optimal performance of the model in both scenarios.

**Adversarial Training**: Adversarial domain adaptation employs a domain discriminator to differentiate between characteristics from the source and target domains. The model is trained to intentionally perplex the domain discriminator, thereby acquiring domain-invariant features. This method enhances the model’s ability to generalize effectively across diverse domains [97].**Feature Alignment:** Feature alignment approaches seek to align the distributions of features between the source and target domains. These objectives can be accomplished using techniques such as Maximum Mean Discrepancy (MMD) and Correlation Alignment (CORAL). Through the process of aligning the distributions, the model is able to acquire knowledge about features that are significant in both domains [98].**Self-Training**: Self-training is a process where the model uses its own predictions on the target domain data as pseudo-labels to train itself again. The iterative method facilitates the model’s steady adaptation to the target domain by refining its predictions and enhancing its performance [99].

#### 5.3.2. Transfer Learning

Transfer learning utilizes acquired knowledge about a given domain (source) and employs it in a different domain (target). This method is advantageous when there is a limited quantity of labeled data in the desired domain. There are several methods to incorporate transfer learning:**Fine-Tuning**: Fine-tuning refers to the process of initially training a model on a large dataset from a certain domain and subsequently refining the model using target domain data. This method capitalizes on the knowledge obtained from pre-training and adapts it to suit the specific target domain [100].**Domain Randomization**: Domain randomization is a simulation approach that involves randomizing certain elements of the simulated environment, such as lighting and texturing, to provide a diverse range of scenarios. This facilitates the model’s acquisition of resilient characteristics that exhibit strong generalization capabilities in real-world scenarios. After being trained on a range of different scenarios, the model is capable of performing efficiently in real-world situations [101].**Multi-Task Learning**: Multi-task learning is the process of training a model on numerous interconnected tasks at the same time. This methodology enables the model to exchange representations between different tasks, hence enhancing its ability to generalize and perform well in the target domain. For instance, a model that has been trained to do both object detection and semantic segmentation can make use of the shared knowledge that exists between these two tasks [102].

#### 5.3.3. Applications and Benefits

Domain adaptation and transfer learning approaches have demonstrated considerable potential in narrowing the disparity between simulated and real-world data in various fields such as autonomous driving, robotics, and computer vision. These methods help in:**Minimizing the Requirement for Abundant Real-World Data**: By utilizing simulated data and transferring expertise to real-world applications, these methods diminish the reliance on substantial quantities of labeled real-world data, which may be expensive and time-consuming to gather.**Improving Model Robustness and Generalization**: Enhancing the model’s domain adaptation and transfer learning improve the ability of models to handle varied real-world settings, hence enhancing their resilience and generalization capabilities.**Accelerating Development and Deployment**: The utilization of easily available simulated data for initial training and its subsequent adaptation for real-world settings allows for the expeditious construction and deployment of models.

#### 5.3.4. Potential Proof of Concept

To illustrate the practicality of these methods, we provide an overview of a potential proof of concept using real-world data experiments:**Data Collection and Preprocessing**: The data collection and preprocessing phase will include collecting data from the KITTI and Waymo Open Datasets, with the aim of including a wide range of driving scenarios and conditions. The data will undergo preprocessing to synchronize and calibrate sensor inputs, such as RADAR and camera data, as described.**Model Training and Adaptation**: The models will undergo initial training using simulated data from the CARLA simulator. Subsequently, domain adaptation and transfer learning methodologies will be utilized to customize these models for real-world data. The training procedure will involve adjusting hyperparameters to enhance performance on various datasets.**Performance Evaluation**: The adapted models will be evaluated using standard metrics such as Mean Absolute Error (MAE), Root Mean Square Error (RMSE), and F1-score. The efficiency of the adaptation approaches will be evaluated by comparing their performance on simulated data with real-world data. The expected results are intended to demonstrate that the adapted models uphold a high level of accuracy and resilience, hence confirming the validity of the suggested methodology.

## 6. Conclusions

An improved multi-object tracking model based on the DeepSORT algorithm was presented in this paper. When fog is present, it can be difficult to detect or track distant or small objects in an autonomous driving environment. As an example of a safety-critical situation, an autonomous driving environment necessitates a higher tracking speed in multi-object tracking models. Object appearance features were extracted using a primitive neural network in the original DeepSORT method. Therefore, the resulting feature map often omits important information about the target being matched with a specific detection. Consequently, identity switches and track failures are more likely to occur when matching objects that look quite similar in the detection frame. Errors in matching can also arise if items of varying sizes are included in the detection frame.

Nevertheless, we used our camera-radar fusion network during the detection phase to increase both the speed with which objects could be detected and the accuracy with which they could be tracked when visibility was extremely low. Instead of using a standard convolutional neural network, we proposed a more robust appearance feature network. We incorporated GhostNet to take on the role of the standard convolutional layers to produce more features and lower computational difficulties and costs while improving tracking speed without reducing the output feature maps. We also included a segmentation module (SM) and gave it the semantic labels from the input frame to enrich the feature maps for the low-level appearance with rich semantic information. Distinguishing between items that appear identical in a noisy background, such as fog, is made easier with the addition of rich semantic information. To deal with the problem of variation in object size on the detection frame, the appearance features were fused at three different depths. Our proposed MOT method performed better than YOLOv5 + DeepSORT, such that under heavy fog conditions, the multi-object tracking accuracy (MOTA) increased by 35.15%, the multi-object tracking precision (MOTP) increased by 32.65%, the speed increased by 37.65%, and identity switches (IDS) decreased by 46.81%.

Future research will focus on enhancing the robustness and real-world applicability of the approach. This includes improving sensor fusion techniques, advancing real-time performance, and leveraging cutting-edge deep learning models and domain adaptation methods. To enhance sensor fusion techniques, advanced algorithms such as transformer-based architectures and graph neural networks will be employed to integrate data from multiple sensors. This integration aims to improve the accuracy and reliability of object detection and tracking systems. Advancing real-time performance involves optimizing algorithms for speed and efficiency, utilizing hardware acceleration, and exploring techniques such as model pruning, quantization, and efficient neural network architectures. Exploring advanced deep learning models, such as transformers, will be executed to improve performance for autonomous vehicle applications. Self-supervised and unsupervised learning techniques will also be explored to reduce the dependency on large-labeled datasets. Implementing domain adaptation techniques, such as adversarial training and domain-invariant feature learning, will bridge the gap between simulated and real-world data. This will improve the generalization capabilities of the models. The immediate next steps involve deploying the proposed system on a real-world autonomous vehicle platform and conducting extensive field tests to evaluate its performance, identify shortcomings, and make necessary adjustments.

## Figures and Tables

**Figure 1 sensors-24-04692-f001:**
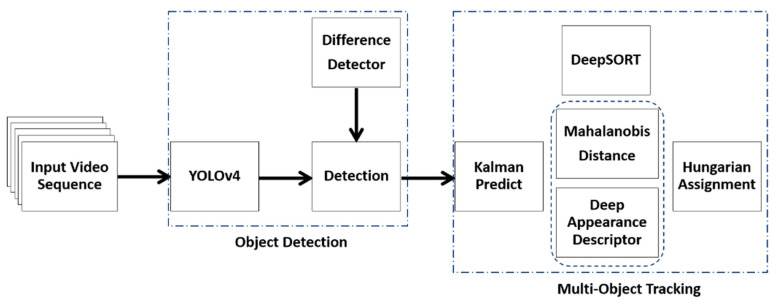
The architecture of the original DeepSORT.

**Figure 2 sensors-24-04692-f002:**
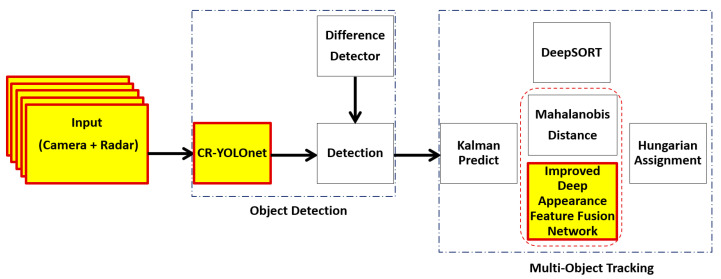
Our improved deep learning-based MOT architecture.

**Figure 3 sensors-24-04692-f003:**
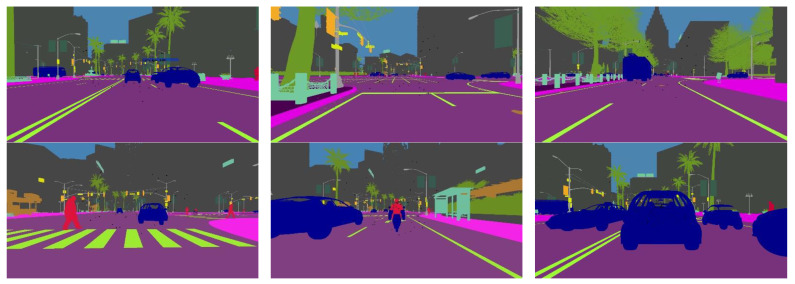
Semantic segmentation showing semantic labels obtained from the semantic segmentation camera in the CARLA simulator.

**Figure 4 sensors-24-04692-f004:**
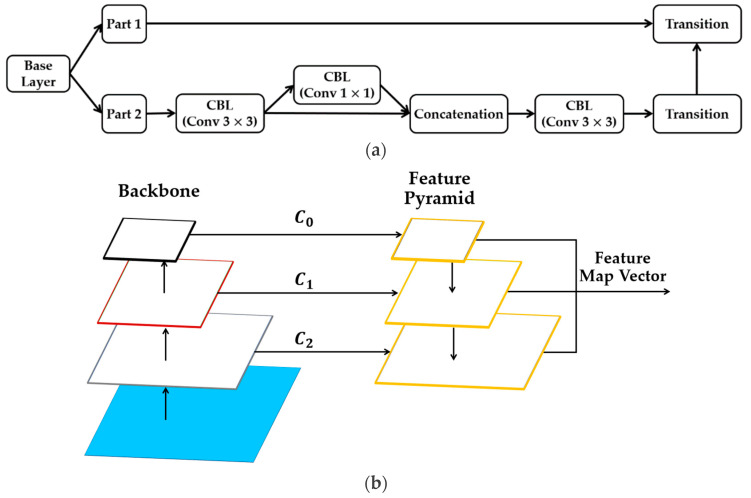
(**a**) the architecture of Cross-Stage Partial Connection, (**b**) the structure of Feature Pyramid Network.

**Figure 5 sensors-24-04692-f005:**
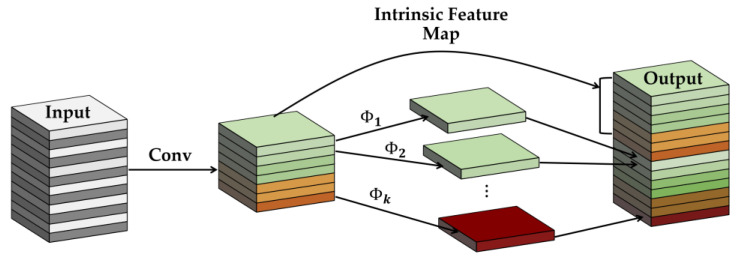
The architecture of the Ghost module.

**Figure 6 sensors-24-04692-f006:**
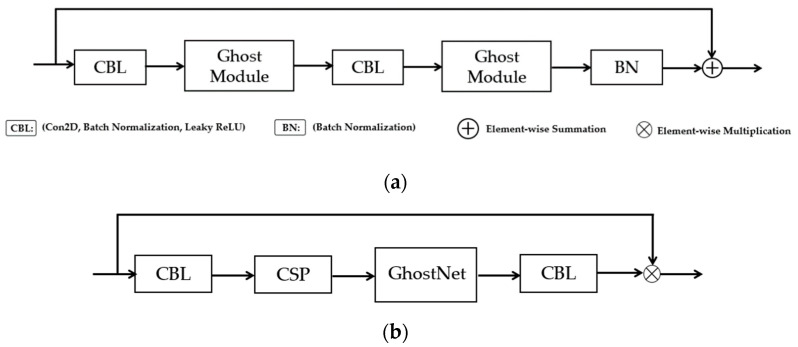
(**a**) Our GhostNet structure; (**b**) the architecture of our Ghost convolution.

**Figure 7 sensors-24-04692-f007:**
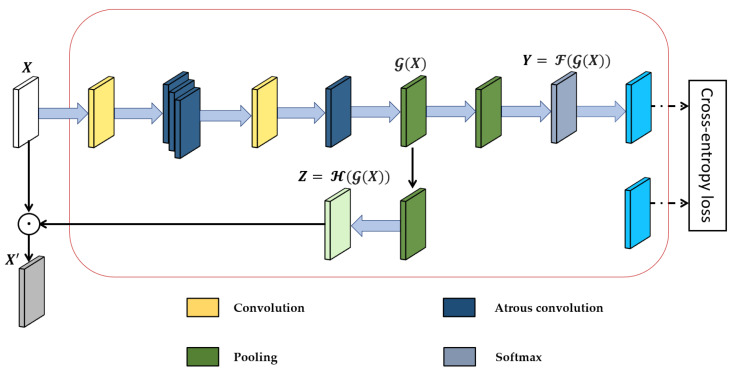
The semantically profound feature map Z is obtained from the primary low-level appearance feature map X (input), the element-wise multiplication of X and Z gives the semantically activated appearance feature map $X^{\prime }$.

**Figure 8 sensors-24-04692-f008:**
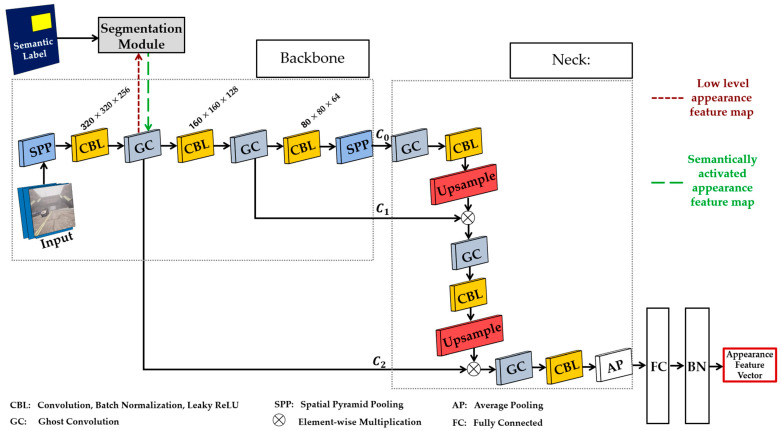
The improved appearance feature extraction network consists of the FPN network that fuses the feature maps $C_{0}$, $C_{1}$, and $C_{2}$ generated by the backbone network at varying depths.

**Figure 9 sensors-24-04692-f009:**
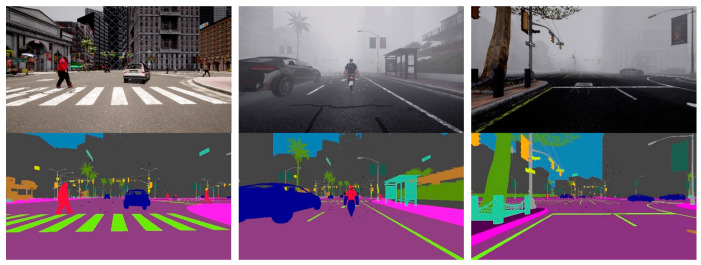
Example of our Carla dataset, including both sunny and foggy conditions, with RGB camera data at the top row and semantic segmentation camera data showing the semantic labels in the second row.

**Figure 10 sensors-24-04692-f010:**
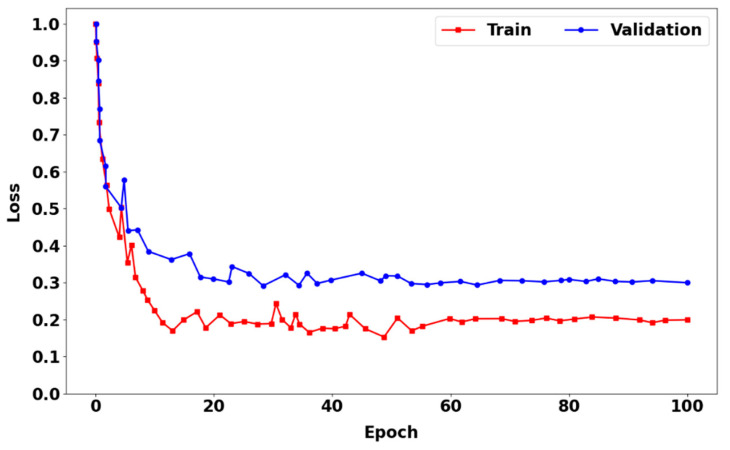
The training and validation loss of our appearance feature extraction network.

**Figure 11 sensors-24-04692-f011:**
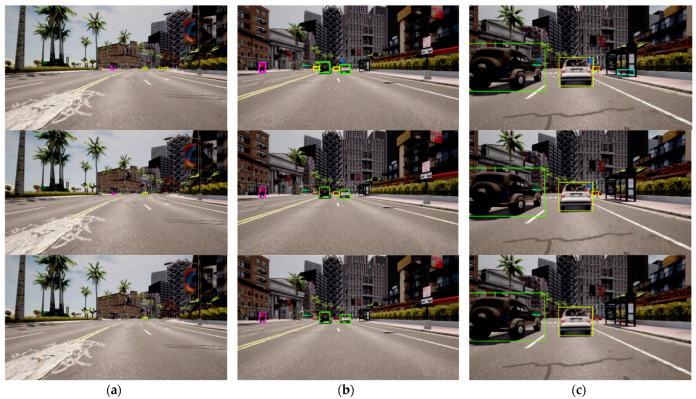
The qualitative results of multi-object tracking in clear weather conditions. Row 1 shows CRYOLO + Ours (CIoU), row 2 shows CRYOLO + Ours (GIoU), row 3 shows CRYOLO + DeepSORT: (**a**) small/distant object, (**b**) medium object, (**c**) large object.

**Figure 12 sensors-24-04692-f012:**
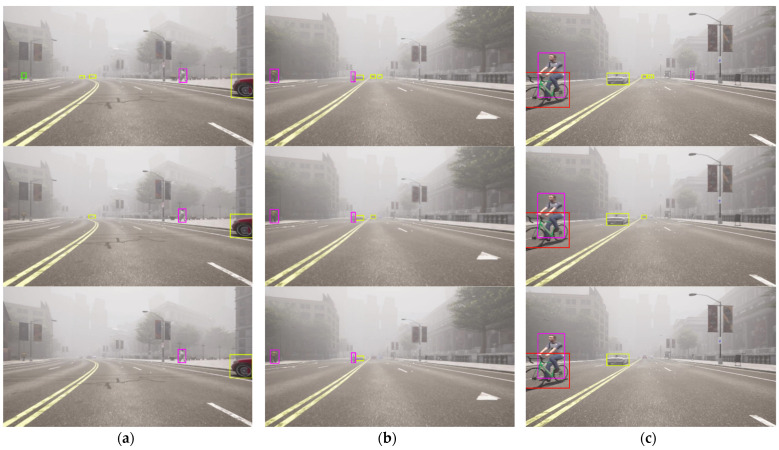
The qualitative results of multi-object tracking in medium fog weather conditions. Row 1 shows CRYOLO + Ours (CIoU), row 2 shows CRYOLO + Ours (GIoU), row 3 shows CRYOLO + DeepSORT: (**a**) small or distant object; (**b**) medium object; (**c**) large object.

**Figure 13 sensors-24-04692-f013:**
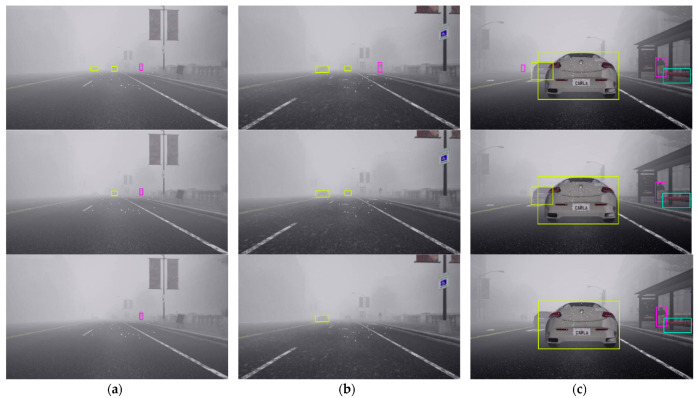
The qualitative results of multi-object tracking in heavy fog weather conditions. Row 1 shows CRYOLO + Ours (CIoU), row 2 shows CRYOLO + Ours (GIoU), row 3 shows CRYOLO + DeepSORT: (**a**) small or distant object; (**b**) medium object; (**c**) large object.

**Table 1 sensors-24-04692-t001:** Comparison of multi-object tracking performance with and without segmentation modules in clear day condition.

Models	Segmentation Module (SM)	MOTA	MOTP	MT	ML	IDS	FP	FN	FPS
CR-YOLO + Ours (CIoU)	WITH	74.86	84.21	43.40%	15.86%	513	5737	1259	68.34
CR-YOLO + Ours (GIoU)	WITH	72.35	83.79	41.80%	17.09%	534	3079	1427	66.09
CR-YOLO + Ours (CIoU)	WITHOUT	71.77	80.10	40.25%	18.06%	598	4765	3882	62.94
CR-YOLO + Ours (GIoU)	WITHOUT	68.07	78.96	35.25%	21.22%	782	5357	4224	58.8

**Table 2 sensors-24-04692-t002:** Comparison of multi-object tracking performance with and without segmentation modules in medium fog condition.

Models	Segmentation Module (SM)	MOTA	MOTP	MT	ML	IDS	FP	FN	FPS
CR-YOLO + Ours (CIoU)	WITH	68.25	79.65	38.77%	17.05%	691	5767	1401	66.15
CR-YOLO + Ours (GIoU)	WITH	66.93	76.99	35.53%	19.24%	723	3628	2183	62.37
CR-YOLO + Ours (CIoU)	WITHOUT	64.93	73.38	33.90%	20.13%	833	4239	5992	59.58
CR-YOLO + Ours (GIoU)	WITHOUT	61.05	70.32	30.16%	23.80%	910	3897	6327	55.95

**Table 3 sensors-24-04692-t003:** Comparison of multi-object tracking performance with and without segmentation modules in heavy fog condition.

Models	Segmentation Module (SM)	MOTA	MOTP	MT	ML	IDS	FP	FN	FPS
CR-YOLO + Ours (CIoU)	WITH	66.14	75.78	36.80%	19.24%	816	7158	5062	64.88
CR-YOLO + Ours (GIoU)	WITH	64.23	73.33	34.02%	21.44%	865	6592	7914	60.42
CR-YOLO + Ours (CIoU)	WITHOUT	60.67	70.54	31.17%	22.79%	1034	10548	5523	56.27
CR-YOLO + Ours (GIoU)	WITHOUT	52.09	66.48	28.76%	26.52%	1193	8349	13626	51.82

**Table 4 sensors-24-04692-t004:** Comparing the performance of the multi-object tracking model with other models in clear day condition.

Models	MOTA	MOTP	MT	ML	IDS	FP	FN	FPS
CR-YOLO + Ours (CIoU) with SM	74.86	84.21	43.40%	15.86%	513	5737	1259	68.34
CR-YOLO + Ours (GIoU) with SM	72.35	83.79	41.80%	17.09%	534	3079	1427	66.09
CR-YOLO + DeepSORT	68.94	78.82	38.11%	20.49%	827	7280	4947	61.24
CR-YOLO + SORT	65.81	72.03	30.94%	22.63%	1129	6945	6740	52.47
YOLOv5 + Ours (CIoU) with SM	67.13	74.91	36.27%	18.29%	841	9394	4235	56.43
YOLOv5 + Ours (GIoU) with SM	66.94	73.58	33.79%	20.45%	867	6651	6685	53.12
YOLOv5 + DeepSORT	63.37	71.41	31.63%	22.95%	893	4476	8708	51.5
YOLOv5 + SORT	60.68	69.48	28.18%	26.43%	1342	6911	10882	44.52

**Table 5 sensors-24-04692-t005:** Comparing the performance of the multi-object tracking model with other models in medium fog conditions.

Models	MOTA	MOTP	MT	ML	IDS	FP	FN	FPS
CR-YOLO + Ours (CIoU) with SM	68.25	79.65	38.77%	17.05%	691	5767	1401	66.15
CR-YOLO + Ours (GIoU) with SM	66.93	76.99	35.53%	19.24%	723	3628	2183	62.37
CR-YOLO + DeepSORT	61.67	71.84	33.31%	22.61%	954	4210	9632	56.8
CR-YOLO + SORT	54.31	64.85	25.60%	36.54%	1685	5877	10184	49.02
YOLOv5 + Ours (CIoU) with SM	63.14	73.42	32.41%	23.09%	976	7394	5185	54.66
YOLOv5 + Ours (GIoU) with SM	60.56	69.25	30.15%	25.03%	1151	6053	7739	52.92
YOLOv5 + DeepSORT	54.98	63.86	27.06%	26.28%	1234	4579	10371	48.71
YOLOv5 + SORT	47.1	58.59	20.72%	29.35%	2425	1706	11125	41.78

**Table 6 sensors-24-04692-t006:** Comparing the performance of multi-object tracking model with other models in heavy fog condition.

Models	MOTA	MOTP	MT	ML	IDS	FP	FN	FPS
CR-YOLO + Ours (CIoU) with SM	66.14	75.78	36.80%	19.24%	816	7158	5062	64.88
CR-YOLO + Ours (GIoU) with SM	64.23	73.33	34.02%	21.44%	865	6592	7914	60.42
CR-YOLO + DeepSORT	56.94	64.23	29.85%	26.56%	1378	6122	8378	52.59
CR-YOLO + SORT	49.67	58.4	21.10%	24.27%	1797	4773	9354	47.9
YOLOv5 + Ours (CIoU) with SM	61.21	70.02	30.34%	21.52%	1213	9612	8941	54.01
YOLOv5 + Ours (GIoU) with SM	55.27	64.74	27.18%	27.88%	1276	2315	9723	51.45
YOLOv5 + DeepSORT	48.94	57.13	24.75%	32.98%	1534	3380	11042	47.14
YOLOv5 + SORT	42.07	53.01	18.87%	38.06%	3305	2911	13185	36.65

## Data Availability

Data are contained within the article.

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
