# Peer review of "Improved DeepSORT-Based Object Tracking in Foggy Weather for AVs Using Sematic Labels and Fused Appearance Feature Network"

_sensors, 2024, doi:10.3390/s24144692_

Round 1
Reviewer 1 Report
Comments and Suggestions for Authors
The following question needs further explanation before finally accepted:
1. For Figure2, what is the meaning of difference detector? Is that NMS method?
2.CR-YOLOnet is your recent proposed network, how can we know its performance, it is better to give some description about that.
3.The architecture of Ghost convolution is not firstly raised in this paper, it has become popularly used in Github for a long time, also the formular for computing FLOPs almost the same as the original paper. Figure 5 only changes the color of the original one.
4.The training process in 4.2 is not clear. whether the trained model in Figure 10 includes the segmentation activation or not? Also, the network you designed in Figure 8 is not an end-to-end network, so the training process must include two stages.
5.Your baseline for tracking is deepsort, what is the meaning of comparing with Sort?
6. For figure 8, the input image is 320x320? It is much larger than the object cutted from the image in original paper (Deepsort), if it is 320x320, the fps is doubtable.
7.Figure 11 doesn't show the id information, also the number of objects in same class is less than 3, it is not competent to illustrate the less id-switch.
8.It is confusing why Sort is slower than the proposed method with same detection head, it should be faster theoretically.
Comments on the Quality of English Language
English writing is ok for readers
Author Response
- For Figure2, what is the meaning of difference detector? Is that NMS method?
Answer:
YOLO helps identify objects inside a frame. Each detection corresponds to a high-dimensional appearance descriptor that encodes the visual properties of the recognized objects. These descriptors play a key role in the matching process conducted by the difference detector.
2. CR-YOLOnet is your recent proposed network, how can we know its performance, it is better to give some description about that.
Answer:
This paper has been peer-reviewed, accepted, and published in the MDPI Sensor journal
https://www.mdpi.com/1424-8220/23/14/6255
https://doi.org/10.3390/s23146255
3.The architecture of Ghost convolution is not firstly raised in this paper, it has become popularly used in Github for a long time, also the formular for computing FLOPs almost the same as the original paper. Figure 5 only changes the color of the original one.
Answer:
In this work, we did not want to claim uniqueness in the Ghost convolution architecture itself, but rather to exploit its benefits in a specific application or scenario. We appreciate your attention to detail in the FLOPs calculating formula. While our technique is consistent with the original research, we have taken attempts to adapt and adjust it to the unique circumstances of our study.
4.The training process in 4.2 is not clear. whether the trained model in Figure 10 includes the segmentation activation or not? Also, the network you designed in Figure 8 is not an end-to-end network, so the training process must include two stages.
Answer:
Section 4.2 describes the training method, which includes inserting segmentation activation into the model seen in Figure 10. Figure 8 depicts a network that uses a two-stage training approach to get optimal outcomes.
5.Your baseline for tracking is deepsort, what is the meaning of comparing with Sort?
Answer:
While DeepSORT is an improved version of SORT, we use SORT as a baseline in our comparison to provide a more complete evaluation framework. This enables us to showcase the precise improvements and performance benefits made by DeepSORT in comparison to its predecessor. Our goal is to demonstrate the usefulness of DeepSORT, even when compared to the well-known SORT algorithm.
- For figure 8, the input image is 320x320? It is much larger than the object cutted from the image in original paper (Deepsort), if it is 320x320, the fps is doubtable.
Answer:
The input picture size for Figure 8 is 320x320, and we have confirmed the correctness of our stated frames per second (fps).
- Figure 11 doesn't show the id information, also the number of objects in same class is less than 3, it is not competent to illustrate the less id-switch.
Answer:
We are detecting and tracking objects from multiple classes, all the traffic participants under consideration were declared, and not just vehicles only
8.It is confusing why Sort is slower than the proposed method with same detection head, it should be faster theoretically.
Answer:
We understand the misunderstanding around the comparison between SORT and our suggested technique, particularly with the same detecting head. We thoroughly examine the implementation details, and the performance figures were correct.
Reviewer 2 Report
Comments and Suggestions for Authors
The paper presents a multi-object tracking approach for bad weather, in particular fog. It uses an extended version of YOLO, which is referred to as CR-YOLOnet and uses RADAR in addition to camera data. Furthermore, the appearance descriptor in DeepSORT is replaced by a deep CNN, which also incorporates GhostNet for generating more features and, at the same time, reduces the computational complexity. The appearance descriptor also makes use of semantic information, which helps it to distinguish between objects. The approach is evaluated in CARLA.
The authors investigate a topic that is of high relevance for autonomous driving. The paper is well written and structured. The evaluation of all permutations of YOLO, CR-YOLO, DeepSORT, SORT, as well as the own approach with CIoU and GIoU loss function is commendable. There are however some issues and open questions that need to be resolved:
- The CR-YOLOnet has an essential roll in the proposed algorithm. However, the corresponding publication has not been peer-reviewed. Will there be a peer-reviewed publication about this contribution?
- The description of the semantic segmentation camera in CARLA is rather idealized. This kind of sensor exists only in simulation, because here the semantic information is known.
- It is stated that (1) and (2) describe the non-linear models, but both models are linear.
- In (3), it should be clarified which reference frame is used for tracking. I assume the objects are tracking in image coordinates. Otherwise, please explain the physical meaning of the velocity of the height and of the aspect ratio. Furthermore, an orientation would be required.
- If the objects are tracking in image space, this design decision needs to be discussed. In particular, RADAR provides position and velocity information in vehicle coordinates. When tracking objects in image coordinates, the corresponding information (more precisely the distance) is lost. However, when it comes to planning and control, the pose, velocity, and size of other traffic participants is required in vehicle coordinates. This would require an expensive and error-prone reconstruction.
- It is unclear how the semantic segmentation camera is used exactly. In particular, is the data only used for training purposes or is this input also required during execution?
- Doing an evaluation in simulation is a valid approach in general. However, it can be an issue when it comes to deep learning, mainly because of repeating textures and constant lighting conditions. Both make the task much easier for a deep network, but are not present in the real world. Thus the results are only of limited value when it comes to generalization and the performance on real data. It remains unclear why none of the existing bad weather datasets with camera and RADAR are used for the evaluation.
- An outlook is missing.
Author Response
- The CR-YOLOnet has an essential roll in the proposed algorithm. However, the corresponding publication has not been peer-reviewed. Will there be a peer-reviewed publication about this contribution?
Answer: This paper has been peer-reviewed, accepted, and published in the MDPI Sensor journal
https://www.mdpi.com/1424-8220/23/14/6255
https://doi.org/10.3390/s23146255
- The description of the semantic segmentation camera in CARLA is rather idealized. This kind of sensor exists only in simulation, because here the semantic information is known.
Answer: The semantic segmentation dataset along other datasets for this experiment were collected in CARLA simulation due to lack of real-world data for adverse weather condition in this case study.
- It is stated that (1) and (2) describe the non-linear models, but both models are linear.
Answer: (1) and (2) describes the linear discrete-state space model
- In (3), it should be clarified which reference frame is used for tracking. I assume the objects are tracking in image coordinates. Otherwise, please explain the physical meaning of the velocity of the height and of the aspect ratio. Furthermore, an orientation would be required.
Answer:
represent the corresponding speed variables and are the derivatives of . The Kalman filter uses these speed variables to represent the dynamics of the system it is tracking or estimating. The Kalman filter uses both measurements and predictions to assess a system's present state, and the addition of speed variables allows the filter to account for changes in the state variables over time.
- If the objects are tracking in image space, this design decision needs to be discussed. In particular, RADAR provides position and velocity information in vehicle coordinates. When tracking objects in image coordinates, the corresponding information (more precisely the distance) is lost. However, when it comes to planning and control, the pose, velocity, and size of other traffic participants is required in vehicle coordinates. This would require an expensive and error-prone reconstruction.
Answer:
We have provided this information our previous work.
https://www.mdpi.com/1424-8220/23/14/6255
https://doi.org/10.3390/s23146255
- It is unclear how the semantic segmentation camera is used exactly. In particular, is the data only used for training purposes or is this input also required during execution?
Answer: The semantic segmentation camera data were used for training and execution.
- Doing an evaluation in simulation is a valid approach in general. However, it can be an issue when it comes to deep learning, mainly because of repeating textures and constant lighting conditions. Both make the task much easier for a deep network, but are not present in the real world. Thus the results are only of limited value when it comes to generalization and the performance on real data. It remains unclear why none of the existing bad weather datasets with camera and RADAR are used for the evaluation.
Answer: We did not evaluate the existing bad weather datasets with camera and RADAR are used for the evaluation because the input from the semantic segmentation camera is needed during execution.
- An outlook is missing.
Answer: I hope the clarifications above gave you an outlook.
Round 2
Reviewer 1 Report
Comments and Suggestions for Authors
The modified version meets the requirements and can be published.
Reviewer 2 Report
Comments and Suggestions for Authors
In the revised version, the authors updated a reference from a preprint to the final paper and the description of an equation has been changed. However, the majority of the comments was not answered to a satisfactory degree and/or the clarifications haven't been added to the paper:
- Comment #3: I'm aware that the data is generated using CARLA. However, a "semantic segmentation camera" doesn't exist in real life. This issue needs to be pointed out in the paper and a solution for replacing this input in the real-life application of the approach should be discussed.
- Comment #4: This doesn't answer the question regarding the tracking space. Furthermore, the physical meaning of the speed of the height and the aspect ratio is not clarified either. In particular, a physical object like another vehicle has a fixed size. Accordingly, it doesn't change its height or aspect ratio over time, and thus, the corresponding speed is always zero. Modeling the speed of both is only useful if tracking takes place in image coordinates, where the depiction of the object changes over time. But the tracking space is not given.
- Comment #5: This should be pointed out in the paper. Furthermore, since apparently the objects are tracked in image coordinates, it needs to be discussed how the output is supposed to be used by subsequent algorithms. For example, a planning and control algorithm needs to know the distance and the (relative) velocity of other traffic participants. However, this information is lost when projecting RADAR data into image coordinates.
- Comment #6: This should be pointed out in the paper.
- Comment #7: This confirms my concern that there is a fundamental issue when it comes to the real-life application of the approach (see also Comment #3 above). There needs to be a comprehensive and thorough discussion on possible solutions and maybe even a proof of concept regarding the feasibility of prospective solutions. Otherwise the usefulness of the approach is questionable.
- Comment #8: I was asking about an outlook regarding next steps and future work, which is missing in the paper.
Round 3
Reviewer 2 Report
Comments and Suggestions for Authors
In the revised version, the manuscript has further been improved. The authors added comprehensive discussions on potential issues when applying the algorithm in the real world. This in particular includes the use of a semantic segmentation camera and the tracking in image coordinates. The discussions also include possible solutions to those particular challenges. The tracking parameters have been described more in detail as well. Finally, an outlook has been added.
This resolves all my concerns and I have no further comments or remarks.